# Reliability and Validity of the Strain Gauge “GSTRENGTH” for Measuring Peak Force in the Isometric Belt Squat at Different Joint Angles

**DOI:** 10.3390/s24103256

**Published:** 2024-05-20

**Authors:** Daniel Varela-Olalla, Carlos Balsalobre-Fernández, Blanca Romero-Moraleda, Sergio L. Jiménez-Sáiz

**Affiliations:** 1Applied Biomechanics and Sports Technology Research Group, Department of Physical Education, Sport and Human Movement, Universidad Autónoma de Madrid, 28049 Madrid, Spain; daniel.varela@uam.es (D.V.-O.); carlos.balsalobre@uam.es (C.B.-F.); blanca.romero@uam.es (B.R.-M.); 2Real Federación Española de Futbol, Las Rozas, 28232 Madrid, Spain; 3Sport Sciences Research Centre, Faculty of Education & Sport Sciences and Interdisciplinary Studies, Universidad Rey Juan Carlos, Fuenlabrada, 28943 Madrid, Spain

**Keywords:** force, isometric training, testing, load cell, performance assessment

## Abstract

Since isometric training is gaining popularity, some devices are being developed to test isometric force as an alternative to the more expensive force plates (FPs); thus, the aim of this study was to test the reliability and validity of “GSTRENGTH” for measuring PF in the isometric belt squat exercise. Five subjects performed 24 contractions at three different knee angles (90°, 105° and 120°) on two occasions (120 total cases). Peak force data were measured using FPs and a strain gauge (SG) and analyzed by Pearson’s product–moment correlation coefficient, ICCs, Cronbach’s alpha, a paired sample *t*-test and Bland–Altman plots. Perfect or almost perfect relationships (r: 0.999–1) were found with an almost perfect or perfect level of agreement (ICCs: 0.992–1; α: 0.998–1). The *t*-test showed significant differences for the raw data but not for the predictions by the equations obtained with the SG values. The Bland–Altman plots, when significant, showed trivial to moderate values for systematic bias in general. In conclusion, “GSTRENGTH” was shown to be a valid alternative to FPs for measuring PF.

## 1. Introduction

Isometric resistance training has been performed for several decades; the main benefits of isometric actions rely on the fact that this type of muscle contraction allows for a greater neural activation, which leads to the activation of a higher number of motor units [1,2], has a reduced metabolic cost compared to concentric muscle actions [3,4], and has the possibility for the greatest voluntary torque to occur in isometric contractions performed at optimum joint angles [5,6]. These characteristics make isometric training a valuable tool for resistance training, as proposed elsewhere [7,8]. 

From intervention studies comparing isometric versus concentric/eccentric training modalities, we can observe that isometric training can promote similar gains in strength than traditional training [9], and its inclusion during traditional resistance training can improve performance both in the lower [10,11] and in the upper body [12] and induce similar improvements in jump performance alongside with greater gains in isometric force compared to plyometric training [13]. 

However, we should consider when programing isometrics that it seems that there exist two types of isometric actions based on the intent of pushing or holding the external weight. The first one is generally referred to as a pushing isometric action, PIMA, or a force task; the second one is known as a yielding isometric action, HIMA, or a position task [14,15,16]. It seems that yielding actions allow for shorter sustained contractions at the same level of force compared to pushing actions [14,15,16], but it is possible that this feature is dependent of the muscle tested [17]. Another important consideration when programing isometric training is the fact that the maximum HIMA force is lower than the maximum PIMA force [18]. 

As stated above, isometric training has been used for many years, but it was not until recently that recommendations based on the adequate percentage of maximum voluntary isometric contraction (MVIC) and contraction times per set and session were proposed [7,8]. For this reason, being able to test and monitor in a precise and reliable way force–time characteristics during isometric training is very relevant in order to ensure proper adaptations and facilitate the training process. Currently, the gold standard for this purpose are force plates (FPs), but they are very expensive and, in some cases, not practical for the daily use. An interesting and cheaper alternative are portable dynamometers and strain gauges (SGs); nevertheless, it is habitual to assume that these devices are adequate for the task even when they have not been validated. Recently, one of these devices, “GSTRENGTH”, was validated against different loads in a weight-bearing manner, showing a perfect relationship between the strain gauge load and the real load [19], but it is still not clear if these results would be replicated if the device was compared against FPs in a more ecologically valid test for sport practitioners, such as isometric exercises. On the other hand, similar devices have been validated for the measurement of peak force (PF) in the isometric knee extension test [20], confirming that they can be useful as an alternative to FPs if these results continue to be repeated.

We hypothesize that new devices like the digital and portable strain gauge “GSTRENGTH” are valid alternatives to FPs. Thus, the objective of the present study is to analyze the reliability and validity of the portable SG “GSTRENGTH” for measuring PF in the isometric belt squat exercise.

## 2. Materials and Methods

### 2.1. Participants

A total of 5 healthy subjects (males = 4, females = 1; age: 25.6 ± 6.31 years, height: 178 ± 9.49 cm, body mass: 75.7 ± 14.52 kg) selected incidentally took part in this study and had at least 1 year of experience in resistance training but not in the isometric belt squat exercise. None of the participants had physical limitations, health problems or injuries at the time of the test. None of the participants were taking drugs, medications or other substances that could alter their physical performance.

### 2.2. Procedures

Subjects attended the laboratory on two occasions separated by 7 days. On the first visit, anthropometric data (height and body mass) and the positions of the belt hooked to a chain for the three knee angles of testing were taken. After that, the subjects performed a warm-up consisting of low-intensity aerobic exercise, body weight squats and isometric squats; finally, subjects performed 12 maximum isometric ramp contractions at 90–105–120 knee degrees (4 contractions per angle; 0 degrees corresponds to full knee flexion) in the isometric belt squat exercise. On the second visit, subjects repeated the same warm-up and testing protocol. The order of the knee angles was randomized for all subjects, and force data were obtained simultaneously with an FP and a portable digital SG. A total of 120 contractions were analyzed. The relationships between the raw values of the SG and FP and the predicted FP values using the SG data were carried out using the whole dataset and grouping the cases by testing sessions and the three different knee angles measured and then dividing the cases by half in a random way (i.e., selecting half of the cases to obtain a prediction equation and testing it against the other half of the data); additionally, a subject-by-subject analysis of the correlation between the SG and FP values was conducted.

### 2.3. Testing Sessions

The subjects arrived at the laboratory at the same time of the day for both sessions. In the first visit, anthropometric measurements were taken using a digital stadiometer with a scale (SECA 2020, SECA, Hamburg, Germany), and then the link of the chain where the belt had to be hooked to ensure that the subject’s knees were as close as possible to 90, 105 and 120 degrees was also set with a self-selected foot position, see Figure 1. Subjects were required to maintain their torso as vertical as possible. In both sessions, the same warm-up was conducted: first, 5 min of low-intensity cycling at a self-selected pace was performed to raise the body temperature; then, 2 sets of 15 body weight squats with a 2 min rest and 2 ramp submaximal isometric contractions of 10” at 105 knee degrees with a 2 min rest were performed. After the warm-up, the subjects performed 4 maximum isometric contractions at 90 knee degrees, 4 maximum isometric contractions at 105 knee degrees and 4 maximum isometric contractions at 120 knee degrees with a 90 s inter-contraction rest in a random order. A total of 24 contractions of 5 s were carried out by each subject. For each contraction, the subjects were asked to maintain the chain without tension until the end of the pre-tension phase of the FP; at this moment, the researcher started the measurement with the SG in the mobile app and asked the participant to start the ramp contraction in a progressive manner by avoiding jerks at the beginning and pushing against the plates for 5 s. Before turning off both devices, the subjects had to stop pushing for a period of 1 s to ensure that the PF (highest value of force recorded during the 5 s contraction) coincided for both instruments and neither instrument was still measuring relevant data while the other was disconnected if any delay occurred during their respective switch-off. PF was obtained with both instruments in order to compare their data for reliability and validity purposes. The subjects were allowed to use a folded towel or jacket to reduce discomfort from the belt.

### 2.4. Instruments

Force plates: All the isometric tests were conducted over a dual FP (2 separate 36.2 × 61 cm plates, Hawkin Dynamics Inc., Westbrook, ME, USA) with a sampling frequency of 1000 Hz placed over a self-made wood platform; data were collected over five seconds via Hawkin Dynamics proprietary software (Hawkin Capture v.8.6.0) installed via an app on a portable tablet (Lenovo Tab P11 Pro, Lenovo®, Beijing, China) that was connected to the FP via Bluetooth. Net PF values were used for analysis.

Digital strain gauge: One digital “GSTRENGTH” SG (Exsurgo Technologies, Ashburn, VA, USA) with a sampling frequency of 80 Hz was attached to the wood platform where the FPs were placed and to one of the ends of the chain; data were collected over five seconds via Exurgo Technologies proprietary software (Exurgo Performance System v.1.6.0) installed via an app on an iPhone 12 that was linked to the SG via Bluetooth. Net PF values were used for analysis.

Both devices were calibrated following the manufacturers’ indications prior to every testing session.

Both devices present net values of force not differentiating by the vertical, anteroposterior or medial–lateral axis.

### 2.5. Statistical Analysis

The normality of the values of PF obtained with the SG and the FP and the values of the predicted FP-PF with the equation obtained in the regression analysis using the SG real values were tested with the Shapiro–Wilk test. The concurrent validity of the SG was tested using Pearson’s product–moment correlation coefficient (r) with 95% confident intervals (CIs) via bootstrapping (n = 1000) with the addition of the root mean square error (RMSE) and determination coefficient (r^2^). To analyze the level of agreement (reliability) between the SG and the FP, the intraclass correlation coefficient (ICC 2.1) and Cronbach’s alpha with a 95%CI were used. In addition, a paired sample *t*-test and Bland–Altman plots were used to identify potential systematic bias by reporting mean bias, standard deviations and the analysis of the regression line of the Bland–Altman plots. The criteria for interpreting the magnitude of the r coefficients and ICCs were trivial (0.00–0.09), small (0.10–0.29), moderate (0.30–0.49), large (0.50–0.69), very large (0.70–0.89), nearly perfect (0.90–0.99) and perfect (1.00) [21]. The level of significance was set at 0.05, and all the analysis were performed using the software package JASP (JASP Team 2023. JASP Version 0.17.1 [Apple Silicon]).

## 3. Results

The values for PF obtained with the FP and the SG and the predicted FP-PF values were distributed normally.

### 3.1. Concurrent Validity

Pearson’s product–moment correlation coefficients showed a significant and perfect relationship for the values obtained with the SG and the FP (r = 1; RMSE: 18.022 N; 95%CI: 0.999–1; *p* < 0.001) and for the predicted FP values using the ones obtained with the SG and the real FP values (r = 1; 95%CI: 0.999–1; *p* < 0.001) when analyzing the whole dataset (Figure 2). When analyzing the data taking the half of the cases in a random order (Figure 3), significant and perfect correlations were also found between the SG and FP (r = 1; RMSE: 17.471–18.811 N; 95%CI: 0.999–1; *p* < 0.001) and when crossing the predicted FP values obtained with the SG and the real FP values (r = 1; 95%CI: 0.999–1; *p* < 0.001). Similar results were also found when analyzing the data based on the testing session (Figure 3) for both sessions and when crossing the predicted FP values obtained with the SG and the real FP values of both sessions (r = 1; RMSE: 18.037–18.117 N; 95%CI: 0.999–1; *p* < 0.001). When analyzing the cases by the three different knee angles tested (Figure 4), significant and almost perfect correlations were found for 90° (n = 40; r = 0.999; RMSE: 16.710 N; 95%CI: 0.998–0.999; *p* < 0.001), 105° (n = 40; r = 0.999; RMSE: 18.289 N; 95%CI: 0.998–1; *p* < 0.001) and 120° (n = 40; r = 1; RMSE: 18.777 N; 95%CI: 0.999–1; *p* < 0.001) and when crossing the predicted equations of the three angles against the real values obtained with the other two (n = 80; r = 1; 95%CI: 0.999–1; *p* < 0.001). Finally, when analyzing the dataset dividing the cases subject by subject, significant and perfect or almost perfect correlations were found for all of them: subject 1 (n = 24; r = 1; RMSE: 11.203 N; 95%CI: 0.999–1; *p* < 0.001), subject 2 (n = 24; r = 1; RMSE: 19.998 N; 95%CI: 0.999–1; *p* < 0.001), subject 3 (n = 24; r = 1; RMSE: 13.798 N; 95%CI: 1–1; *p* < 0.001), subject 4 (n = 24; r = 1; RMSE: 19.566 N; 95%CI: 0.998–1; *p* < 0.001) and subject 5 (n = 24; r = 0.999; RMSE: 16.608 N; 95%CI: 0.997–0.999).

### 3.2. Reliability of the SG Compared to the FP

A very high level of agreement was revealed by the ICCs and Cronbach’s alpha values obtained for all the analyses performed: general equation (ICC: 0.999, 95%CI: 0.98–1; α: 1, 95%CI: 1–1); random analysis (ICC: 1, 95%CI: 0.999–1; α: 1, 95%CI: 1–1); session-b-session analysis (ICC: 0.999–1, 95%CI: 0.999–1; α: 1, 95%CI: 1–1); 90° knee angle (ICC: 0.992, 95%CI: 0.945–0.997; α: 0.998, 95%CI: 0.996–0.999), 105° knee angle (ICC: 0.997, 95%CI: 0.892–0.999; α: 1, 95%CI: 0.999–1) and 120° knee angle (ICC: 0.999, 95%CI: 0.972–1; α: 1, 95%CI: 1–1); subject 1 (ICC: 0.999, 95%CI: 0.978–1; α: 1, 95%CI: 1–1), subject 2 (ICC: 0.996, 95%CI: 0.98–0.998; α: 0.998, 95%CI: 0.997–0.999), subject 3 (ICC: 0.999, 95%CI: 0.973–1; α: 1, 95%CI: 0.999–1), subject 4 (ICC: 0.998, 95%CI: 0.964–0.999; α: 1, 95%CI: 1–1) and subject 5 (ICC: 0.994, 95%CI: 0.679–0.999; α: 0.999, 95%CI: 0.998–1).

The paired sample *t*-test showed systematic bias when comparing the values obtained with the SG and the FP for the overall data, the random analysis, the test-by-test analysis, the three different knee angles and for all the subjects (*p* ≤ 0.002). When comparing the predicted FP values versus the real FP values, no systematic bias was observed for the overall dataset (*p* = 0.895) and the predicted data crossed in a random way (*p* = 0.543–0.689) or based on the testing sessions (*p* = 0.173–0.211); however, all predicted data crossed by the knee angle showed systematic bias (90°: *p* = 0.038; 105°: *p* = 0.003; 120°: *p* < 0.001). Table 1 and Table 2 show the values for the absolute and relative bias with their respective limits of agreement.

The Bland–Altman plots showed a small significant relationship (r^2^: 0.196, *p* < 0.001) for the absolute differences between the SG and FP and a trivial non-significant relationship for the relative differences (r^2^: 0.002, *p* = 0.695) (Figure 5). When splitting the cases by half in a random way, moderate relationships were observed for the absolute differences (r^2^: 0.144–0.239; *p* ≤ 0.003) and trivial relationships for the relative differences (r^2^: 0.001; *p* = 0.792–0.804) (Figure 5). On the other hand, splitting the cases by test session showed small correlations for the absolute differences (r^2^: 0.132–0.263, *p* ≤ 0.004) and trivial correlations for the relative differences (r^2^: 0.008–0.022, *p* = 0.26–0.491) (Figure 5). In addition, trivial and non-significant relationships were observed for the absolute (r^2^: 0.001–0.009; *p* = 0.339–0.566) and relative (r^2^: 0.001–0.04; *p* = 0.123–0.794) differences between the predicted values and real FP values crossed in a random way and based on testing sessions (Figure 5). Trivial non-significant relationships were also observed for the absolute and relative differences in PF at 90° and 105° knee angles (r^2^: 0–0.04; *p* = 0.207–0.996); for 120°, a moderate significant relationship (r^2^: 0.301; *p* < 0.001) was found for the absolute differences and a trivial non-significant relationship was found for the relative differences (r^2^: 0.04; *p* = 0.201) (Figure 6). For the predicted values, crossing the equations of the three angles against the real values obtained with the other two Bland–Altman plots showed trivial to moderate relationships for the absolute differences (r^2^: 0.076–0.499; *p* ≤ 0.013) and trivial to small relationships for the relative differences (r^2^: 0.085–0.224; *p* ≤ 0.009) (Figure 6). Finally, when the cases were analyzed in a subject-by-subject way, small to large correlations were found for the absolute differences (subject 1, r^2^: 0.683, *p* < 0.001; subject 2, r^2^: 0.347, *p* = 0.002; subject 3, r^2^: 0.391, *p* = 0.001; subject 4, r^2^: 0.149, *p* = 0.063; subject 5, r^2^: 0.039, *p* = 0.357) and trivial to small correlations for the relative differences (subject 1, r^2^: 0.247, *p* = 0.013; subject 2, r^2^: 0.18, *p* = 0.039; subject 3, r^2^: 0.032, *p* = 0.401; subject 4, r^2^: 0.002, *p* = 0.82; subject 5, r^2^: 0.27, *p* = 0.009).

## 4. Discussion

“GSTRENGTH” was found to be reliable and valid in measuring PF in the isometric belt squat exercise. To our knowledge, this is the first study analyzing the reliability and validity of the strain gauge “GSTRENGTH” for measuring PF in the isometric belt squat exercise; however, it has been recently validated by Ripley and McMahon [19], showing a perfect relationship between the SG load and the real load hanging of it. Our results coincide with those of these authors since all the analysis performed for testing the concurrent validity between the instruments showed perfect or almost perfect correlations between the values of the SG and the FP (r: 0.999–1) with a very narrow CI via bootstrapping analysis (0.998–1). In addition, the slopes of the regression lines (s = 0.95–1.03) were very close to the regression line (y = x), indicating that the results of both devices were practically identical.

For the reliability analysis, a very high perfect or an almost perfect level of agreement was found for the ICCs (0.992–1) and Cronbach’s alpha (α: 0.998–1); nonetheless, the paired sample *t*-test showed a systematic bias for all the analyses comparing the real values obtained with the SG versus the FP, by which the values obtained with the SG were slightly higher than those of the FP. No systematic biases were observed when comparing the values predicted with the equations obtained with the SG data versus the real values of the FP, with the exception of the prediction equations based on the different knee angles tested. However, the mean absolute and relative values of the differences between the instruments were under 30 N and 5%, respectively, for all the analyses.

Our results indicate that the “GSTRENGTH” SG is a valid option for measuring PF, at least in the isometric belt squat exercise, and coincide with those of previous studies validating similar devices [19,20]. Overall, studies testing the validity of this device, this paper and the one by Ripley and McMahon [19], showed perfect relationships, reinforcing the use of the raw data of the SG or applying the correction factor from the regression equations obtained from it. However, a couple of factors should be considered based on our data. First, although the correlations were still almost perfect (r ≥ 0.999), the analysis performed splitting the data in a subject-by-subject way showed a wider CI for the ICCs and also small to large correlations for the absolute differences in the Bland–Altman plots (r^2^: 0.039–0.683). Of note, the relative differences showed much lower correlation values (r^2^: 0.002–0.27), and it is also important to note that the subjects with the higher determination coefficients in the Bland–Altman plots (subjects 1 and 3) were also the ones with greater values for the Pearson correlation coefficients (r: 1) with the lower RMSEs (11.203–13.978 N; whole dataset RMSE: 18.022 N); in addition, greater reductions in the r^2^ of the Bland–Altman determination coefficients were shown after transforming the absolute to relative values. On the other hand, splitting cases by the three knee angles tested also showed higher r^2^ values for the Bland–Altman plots compared with the whole data analysis when crossing the prediction equations of the three angles against the real values obtained with the other two (see Section 3). First, this could be due to the fact that the subjects were not used to training with the isometric belt squat, and secondly to the fact that different degrees of knee flexion (i.e., 90° vs. 120° knee angle) make it more difficult to maintain the initial position without jerking or reducing to the minimum the initial force applied to the FP, which could influence the initial force measured during the stance phase that will be subtracted to calculate the net PF and increase the difference between both devices. This first possibility is in line with the results of Juneau et al. [20], who showed that the variability was reduced from session one to session three, indicating that allowing the subject to familiarize with the task measured could be important in order to increase the reliability of the data; however, in our study, no clear differences were observed between the testing sessions for the overall dataset, but an increase from 0.132 to 0.263 and a reduction from 0.022 to 0.008 in the Bland–Altman plots for the absolute and relative differences were observed. The influence of the knee angle was also analyzed by Juneau et al. [20], and their results showed greater stability for 90° of knee flexion versus 60° (0° in their study means that the knee is completely extended), but our results showed a reduction in the r^2^ values of the Bland–Altman plots for both the absolute and relative differences from 90° to 105° and then an increase in the absolute differences at 120°; in addition, the ICCs were greater for 105° and 120° compared to 90° (see Section 3). This later comparison between our three knee angles and the results of Juneau et al. [20] could indicate that different exercises could have different optimal angles and set-ups for measuring PF with an SG since they used the isometric knee extension and we tested the isometric belt squat.

When analyzing the whole dataset or splitting the cases randomly or by testing session, the influence of the knee angle and the subject-by-subject variability in the results was diminished, indicating that the “GSTRENGTH” SG can be used by practitioners for measuring PF, and for research purposes, using the correction factors presented here increases the reliability of data, making it similar to those obtained with the FP without a risk of bias.

The main limitations in the present study are as follows: (1) the low number of subjects (n = 5) that performed the test, which did not allow us to make clear statements about the influence of subjects’ experience, sex, etc., on their individual results and test its influence on the overall data; (2) the impossibility to completely control the force measured during the stance phase for the FP when changing the knee angle and therefore its effect on the difference between devices; (3) when analyzing the data in a subject-by-subject or angle-by-angle way, the number of cases used in the correlations was reduced to less than half of the total sample, and this could influence the results; (4) the fact that only PF was tested and other kinetic variables like RFD or impulse will be of interest, as the literature shows that other SG devices present greater variability for these parameters [20,22,23]; (5) padding via a folded towel or jacket was required due to discomfort caused by the edges of the belt, which could have impacted the results, affecting the methodological standardization; (6) only the isometric belt squat exercise was tested here, and other exercises with different set-ups and constraints should be tested in order to generalize our results; (7) only maximum contractions were tested here, so the range of force values is limited, and intermediate or extreme minimal values could present different results.

## 5. Conclusions

In conclusion, the “GSTRENGTH” SG was proven to be a valid and accurate alternative to the more expensive FP for measuring PF during the isometric belt squat exercise, showing a trivial to small systematic bias in general for the raw data (absolute mean bias: 24.31 ± 20.16 N; relative mean bias: 1.66 ± 1.52%), but not when the data are converted using the correction factors presented with our regression models when the whole dataset is analyzed or when divided by sessions or randomly. However, the influence of the subjects’ ability to perform the task and the use of different joint angles should be accounted for in each user since they seem to increase the risk of bias mainly by hindering the standardization of the set-up due to changes in pretension during the stance phase of the FP in this study. However, the total number of cases for the subject-by-subject and angle-by-angle analyses was reduced to less than half of the total sample, which could also affect the results, and therefore further research is warranted for these aspects. Finally, future research is also warranted in order to test the reliability and validity of this device for measuring RFD and also for testing the SG device for measuring force values different from the maximum peak force.

## Figures and Tables

**Figure 1 sensors-24-03256-f001:**
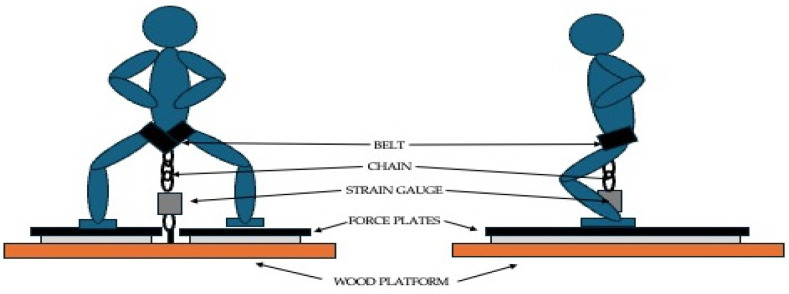
Example of the configuration and positioning for an isometric squat. The illustration aims to represent 90 degrees of knee flexion, and a similar set-up was employed for 105 and 120 degrees.

**Figure 2 sensors-24-03256-f002:**
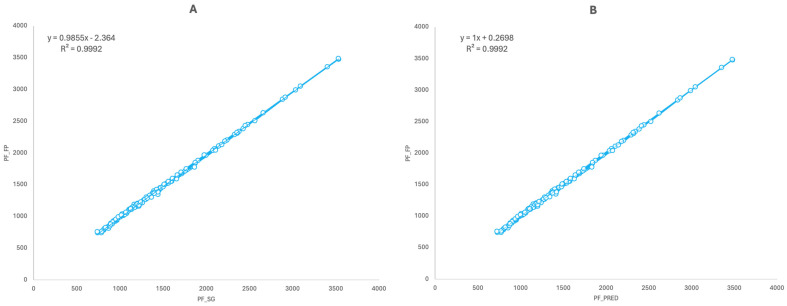
(**A**) Concurrent validity between the SG and FP for the whole dataset (n = 120). (**B**) Relationship between the actual values of the FP and the predicted values using the regression equation shown in (**A**) (n = 120). Data are presented in N.

**Figure 3 sensors-24-03256-f003:**
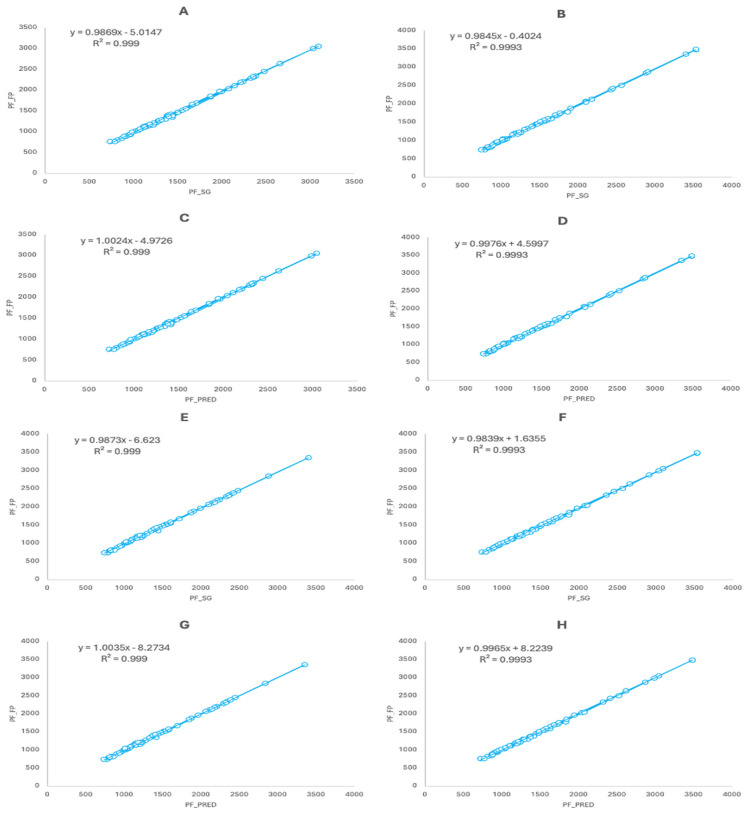
(**A**) Concurrent validity between the SG and FP for the first half of the data selected randomly (n = 60). (**B**) Concurrent validity between the SG and FP for the second half of the data selected randomly (n = 60). (**C**) Relationship between the actual values of the FP selected for B and the predicted values using the regression equation shown in (**A**) (n = 60). (**D**) Relationship between the actual values of the FP selected for (**A**) and the predicted values using the regression equation shown in (**B**) (n = 60). (**E**) Concurrent validity between the SG and FP for the first testing session (n = 60). (**F**) Concurrent validity between the SG and FP for the second testing session (n = 60). (**G**) Relationship between the actual values of the FP for the second testing session and the predicted values using the regression equation shown in (**E**) (n = 60). (**H**) Relationship between the actual values of the FP for the first testing session and the predicted values using the regression equation shown in (**F**) (n = 60). Data are presented in N.

**Figure 4 sensors-24-03256-f004:**
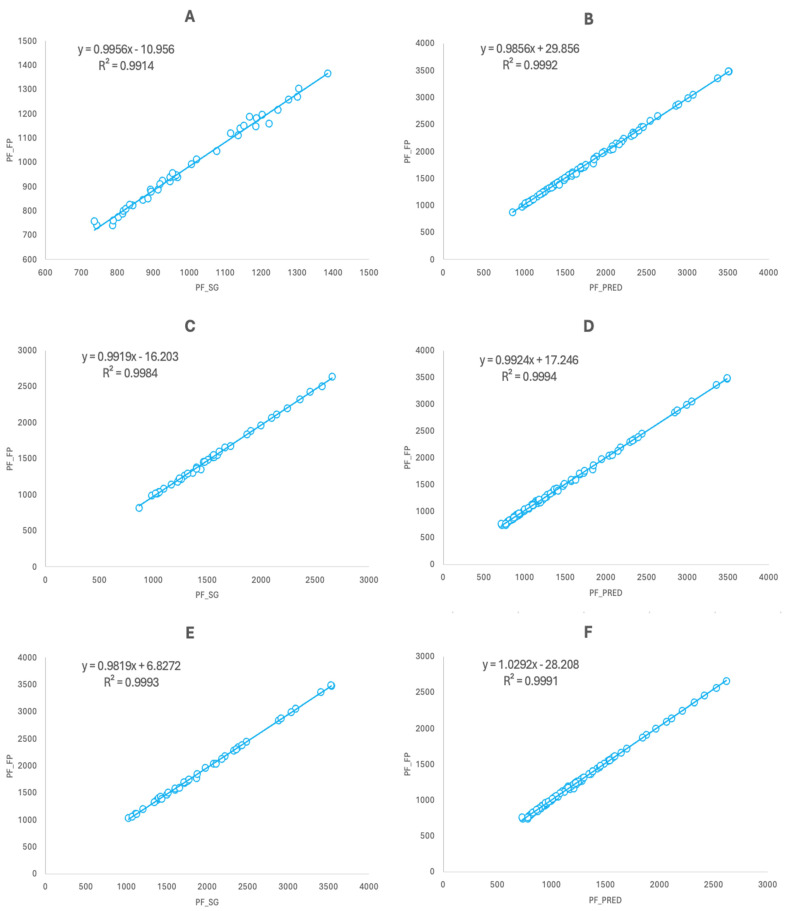
(**A**) Concurrent validity between the SG and FP for the 90° knee angle (n = 40). (**B**) Relationship between the actual values of the FP at 105° and 120° and the predicted values using the regression equation shown in (**A**) (n = 80). (**C**) Concurrent validity between the SG and FP for the 105° knee angle (n = 40). (**D**) Relationship between the actual values of the FP at 90° and 120° and the predicted values using the regression equation shown in (**C**) (n = 80). (**E**) Concurrent validity between the SG and FP for the 120° knee angle (n = 40). (**F**) Relationship between the actual values of the FP at 90° and 105° and the predicted values using the regression equation shown in (**E**) (n = 80). Data are presented in N.

**Figure 5 sensors-24-03256-f005:**
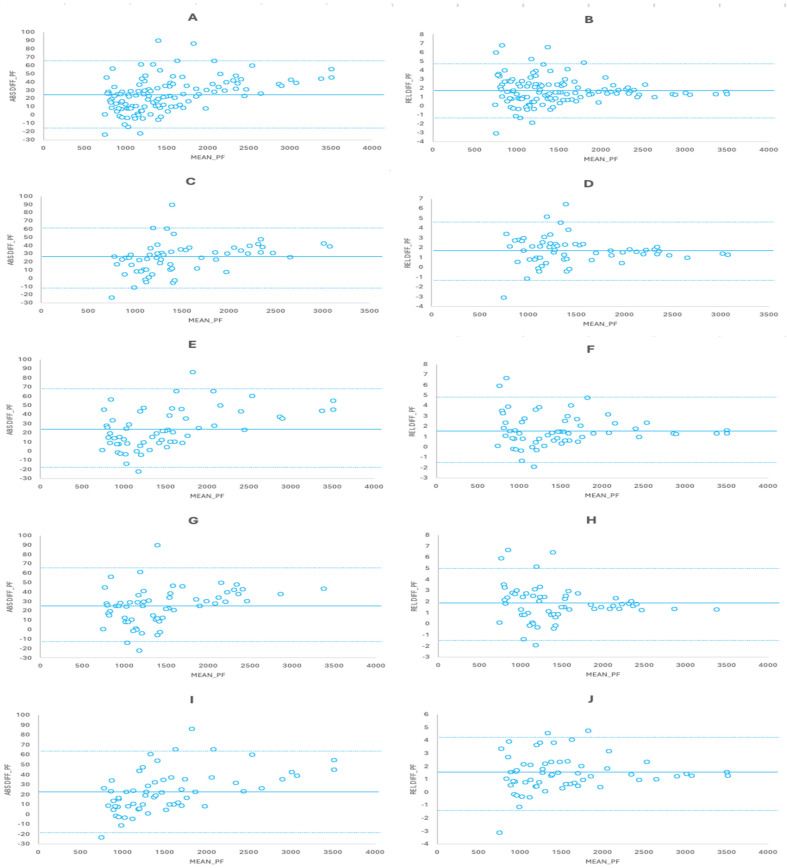
Bland–Altman plots for the absolute and relative differences between the values obtained with the SG and the FP. (**A**) Absolute difference for the whole dataset (n = 120). (**B**) Relative differences for the whole dataset (n = 120). (**C**) Absolute difference for the first half of the data selected randomly (n = 60). (**D**) Relative differences for the first half of the data selected randomly (n = 60). (**E**) Absolute difference for the second half of the data selected randomly (n = 60). (**F**) Relative differences for the second half of the data selected randomly (n = 60). (**G**) Absolute difference for the first testing session (n = 60). (**H**) Relative differences for the first testing session (n = 60). (**I**) Absolute difference for the second testing session (n = 60). (**J**) Relative differences for the second testing session (n = 60). Continuous line represents the mean bias and dotted lines represent the limits of agreement. Data for absolute differences are presented in N and for relative differences as percentages.

**Figure 6 sensors-24-03256-f006:**
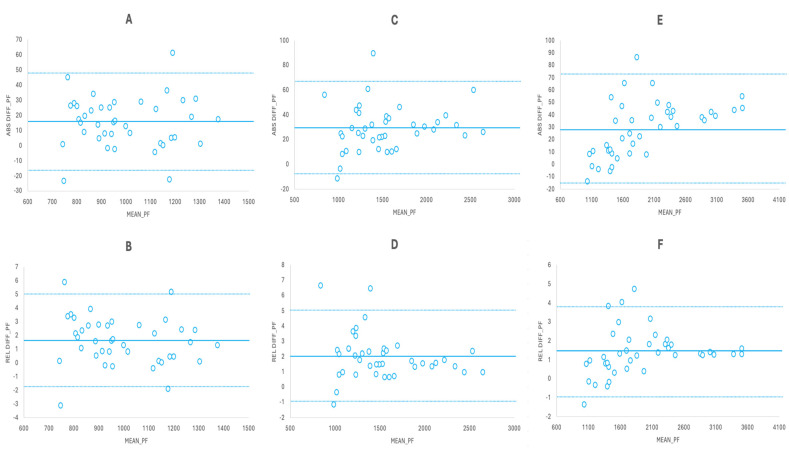
Bland–Altman plots for the absolute and relative differences between the values obtained with the SG and the FP. (**A**) Absolute difference for the 90° knee angle (n = 40). (**B**) Relative difference for the 90° knee angle (n = 40). (**C**) Absolute difference for the 105° knee angle (n = 40). (**D**) Relative difference for the 105° knee angle (n = 40). (**E**) Absolute difference for the 120° knee angle (n = 40). (**F**) Relative difference for the 120° knee angle (n = 40). Continuous line represents the mean bias and dotted lines represent the limits of agreement. Data for absolute differences are presented in N and for relative differences as percentages.

**Table 1 sensors-24-03256-t001:** Absolute bias for mean differences between the SG and the FP.

Analysis	Mean Bias	Mean 95%CI	Upper LoA	Upper LoA 95%CI	Lower LoA	Lower LoA 95%CI
General	24.31 ± 20.16 N	20.66 N–27.95 N	63.81 N	57.5 N–70.12 N	−15.2 N	−21.51 N–(−8.89 N)
Random division 1	24.7 ± 18.84 N	19.83 N–29.57 N	61.63 N	53.2 N–70.06 N	−12.23 N	−20.66 N–(−3.8 N)
Random division 2	23.92 ± 21.54 N	18.35 N–29.48 N	66.13 N	56.5 N–75.77 N	−18.3 N	−27.94 N–(−8.67 N)
Test 1	25.27 ± 19.41 N	20.23 N–30.28 N	63.31 N	54.62 N–71.99 N	−12.77 N	−21.45 N–(−4.08 N)
Test 2	23.35 ± 21 N	17.92 N–28.77 N	64.5 N	55.11 N–73.9 N	−17.81 N	−27.21 N–(−8.42 N)
90°	15.35 ± 16.51 N	10.07 N–20.63 N	47.72 N	38.57 N–56.86 N	−17.02 N	−26.16 N–(−7.87 N)
105°	28.83 ± 18.43 N	22.94 N–34.73 N	64.96 N	54.75 N–75.17 N	−7.3 N	−17.51 N–2.91 N
120°	28.74 ± 22.44 N	21.56 N–35.92 N	72.73 N	60.3 N–85.16 N	−15.25 N	−27.68 N–(−2.82 N)
Subject 1	26.53 ± 19.65 N	18.23 N–34.82 N	65.04 N	50.67 N–79.41 N	−11.99 N	−26.36 N–2.39 N
Subject 2	17.98 ± 24.76 N	7.52 N–28.44 N	66.51 N	48.4 N–84.62 N	−30.55 N	−48.66 N–(−12.44 N)
Subject 3	21.52 ± 17.47 N	14.14 N–28.9 N	55.77 N	42.99 N–68.55 N	−12.73 N	−25.51 N–0.05 N
Subject 4	25.71 ± 20.9 N	16.89 N–34.54 N	66.69 N	51.4 N–81.97 N	−15.26 N	−30.55 N–0.03 N
Subject 5	29.8 ± 16.49 N	22.84 N–36.76 N	62.11 N	50.05 N–74.17 N	−2.51 N	−14.57 N–9.55 N

Data for mean bias are presented as mean ± SD. Mean 95%CI: 95% confident intervals for mean difference; upper LoA: upper limit of agreement; upper LoA 95%CI: 95% confident intervals for upper limit of agreement; lower LoA: lower limit of agreement; lower LoA 95%CI: 95% confident intervals for lower limit of agreement.

**Table 2 sensors-24-03256-t002:** Relative bias for mean differences between the SG and the FP.

Analysis	Mean Bias	Mean 95%CI	Upper LoA	Upper LoA 95%CI	Lower LoA	Lower LoA 95%CI
General	1.66 ± 1.52%	1.38–1.93%	4.64%	4.17–5.12%	−1.33%	−1.81–(−0.85)%
Random division 1	1.68 ± 1.45%	1.3–2.05%	4.53%	3.88–5.18%	−1.17%	−1.83–(−0.52)%
Random division 2	1.61 ± 1.57%	1.21–2.02	4.69%	3.99–5.39%	−1.47%	−2.17–(−0.76)%
Test 1	1.82 ± 1.6%	1.41–2.24%	4.96%	4.24–5.67%	−1.31%	−2.02–(−0.59)%
Test 2	1.46 ± 1.4%	1.1–1.83%	4.21%	3.59–4.84%	−1.28%	−1.91–(−0.66)%
90°	1.59 ± 1.74%	1.03–2.15%	5%	4.04–5.97%	−1.82%	−2.79–(−0.86)%
105°	1.96 ± 1.51%	1.48–2.44%	4.92%	4.08–5.75%	−1%	−1.83–(−0.16)%
120°	1.38 ± 1.21%	1–1.77%	3.75%	3.08–4.42%	−0.98%	−1.65–(−0.31)%
Subject 1	1.07 ± 0.88%	0.69–1.44%	2.8%	2.16–3.45%	−0.67%	−1.31–(−0.02)%
Subject 2	1.32 ± 1.77%	0.57–2.07%	4.79%	3.49–6.08%	−2.15%	−3.44–(−0.85)%
Subject 3	1.48 ± 1.14%	0.99–1.96%	3.72%	2.88–4.56%	−0.76%	−1.6–(0.07)%
Subject 4	1.72 ± 1.45%	1.11–2.33.%	4.56%	3.5–5.62%	−1.12%	−2.18–(−0.06)%
Subject 5	2.63 ± 1.71%	1.91–3.36%	5.99%	4.74–7.24%	−0.72%	−1.97–0.53%

Data for mean bias are presented as mean ± SD. Mean 95%CI: 95% confident intervals for mean difference; upper LoA: upper limit of agreement; upper LoA 95%CI: 95% confident intervals for upper limit of agreement; lower LoA: lower limit of agreement; lower LoA 95%CI: 95% confident intervals for lower limit of agreement.

## Data Availability

Data are contained within the article.

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
