# Peer review of "Reliability and Validity of the Strain Gauge “GSTRENGTH” for Measuring Peak Force in the Isometric Belt Squat at Different Joint Angles"

_sensors, 2024, doi:10.3390/s24103256_

Round 1
Reviewer 1 Report
Comments and Suggestions for Authors
Dear Authors,
Thank you for your contribution to understanding the use of the "GSTRENGTH" device in measuring isometric force. Your study presents a robust methodology, and your findings offer valuable insights into the reliability and validity of this instrument.
However, I would like to suggest a few improvements to enhance the relevance of your work:
Detailed Illustrations: Including representative figures of the "GSTRENGTH" device's configuration and positioning during experiments would clarify its use and help readers better understand the setup.
Multi-Axial Analysis: It is unclear if the measurement instruments function solely on one axis. Expanding the analysis to include anteroposterior and lateral dimensions could enrich the understanding of squat movement dynamics and provide a more comprehensive perspective on athletic performance.
Calibration of Instruments: Confirming the calibration of the measurement instruments would strengthen the validity of the results, especially under extreme load conditions.
Amplitude of Force Values: Discussing the amplitude of observed isometric force values would be useful to assess the consistency of measurements between the two systems under varying load conditions.
These suggestions aim to maximize the impact of your research by deepening the analysis and improving the clarity of the presented information.
Thank you again for your work.
Kind regards,
You mention:
p1. l41-48. It would be pertinent to include a discussion on concentric/eccentric stato-dynamic effort.
p2. l50... Regarding the use of this type of training: although not recognized as "isometric" in modern terms, various forms of static exercises were commonly used in traditional training practices of several cultures before the 20th century. It was during the 1950s-1960s that researchers like Hettinger and Muller in Germany conducted systematic studies on the effects of isometric muscle contraction. They established scientific bases indicating that isometric training could increase muscle strength, documenting that regular isometric contractions could improve muscle strength by up to 5% per week.
p2. l50... I agree with the importance of tempo in weight training.
Materials and Methods:
It is regrettable that the "GSTRENGTH" measurement instrument was not presented with precision, lacking a visual support of its configuration and positioning on the subjects. One or two illustrative figures would be helpful and enlightening for the protocol.
p3 l 119-129. Have the measurement instruments been calibrated?
p5 & p6. Figures 2 and 3 are complex to evaluate although they are supposed to be adequate.
Note: The analysis is limited to a single axis! These are not triaxial measurement instruments. In the execution of a squat, its analysis, learning, and optimization, it could be crucial to be able to record anteroposterior and lateral values.
Results:
The document does not seem to mention the amplitude of the isometric force values developed by the subjects during the experiments. This would be necessary to verify if, under extreme load constraints, the comparison of values between the two systems remains coherent and statistically validated. It is possible that inconsistencies exist for these two systems for extreme minimal and/or maximal values.
Conclusion:
You could have tested the concordance of the values of the two measurement instruments through a static (isometric) maximum load constraint, using a system with a cable passing through a pulley, the other end being attached to the "GSTRENGTH" itself attached to the force platform.
References:
Please standardize the following references:
ABBOTT, B.C.; WILKIE, D.R.
"VALIDATION OF A COMMERCIALLY AVAILABLE STRAIN GAUGE AGAINST A SERIES OF KNOWN LOADS USING A SHORT TIME APPROACH.”
Author Response
REPLY TO REVIEWER
SENSORS
---------------------------
07 May 2024
Dear reviewer.
The authors of the manuscript want to thank you for your time and effort, as well as for your recommendations that for sure have helped us to improve the quality of our work.
In the following lines we specified the changes pertinent to the suggestions with the manuscript lines in which they should appear, in the manuscript changes are resalted in red to help the next revision. If the authors thinks that any suggestion is not suitable or appropriate a reasoned explanation will be presented.
Reviewer 1 comments:
p1. l41-48. It would be pertinent to include a discussion on concentric/eccentric stato-dynamic effort.
p2. l50... Regarding the use of this type of training: although not recognized as "isometric" in modern terms, various forms of static exercises were commonly used in traditional training practices of several cultures before the 20th century. It was during the 1950s-1960s that researchers like Hettinger and Muller in Germany conducted systematic studies on the effects of isometric muscle contraction. They established scientific bases indicating that isometric training could increase muscle strength, documenting that regular isometric contractions could improve muscle strength by up to 5% per week.
p2. l50... I agree with the importance of tempo in weight training.
Author´s response: Done. We want to thank the reviewer for the recommendations about possible clarifications about different muscular efforts and/or contraction types in the introduction of the article. We have added a specification (ln. 36-37) stating that second paragraph of the introduction (ln. 36-41) refers to studies comparing dynamic efforts versus static efforts. However, we think that a brief discussion about intervention studies comparing concentric/eccentric or plyometric versus isometric training has already been presented and should be enough as the main objective of the paper is to test the reliability and validity of the strain gauge and not compare different training modalities. If the readers need to go deeper on this last topic they can go through the cited references (i.e. ref 7,8).
Reviewer 1 comments:
Detailed Illustrations: Including representative figures of the "GSTRENGTH" device's configuration and positioning during experiments would clarify its use and help readers better understand the setup.
It is regrettable that the "GSTRENGTH" measurement instrument was not presented with precision, lacking a visual support of its configuration and positioning on the subjects. One or two illustrative figures would be helpful and enlightening for the protocol.
Author´s response: Done. A detailed illustration of the set up for configuration and positioning of both devices is added in the “Material and Methods” section (lines 134-136), consequently the number of figures goes from 5 to 6 and the rest of illustration have been renumbered (lines 160, 167-168, 171, 174, 187, 198, 236, 239, 241, 244, 248, 252, 259, 270).
Reviewer 1 comments:
- p5 & p6. Figures 2 and 3 are complex to evaluate although they are supposed
to be adequate.
Authors response: Done. Figures 2 and 3 have been reattached (lines 186, 197) to the manuscript with better quality. Note that due to the addition of one more figure in response to your previous comment this figures, and the following ones have been renumbered. At the end of this response the new figures are also attached at full page size so that they can be seen more clearly in case they continue to lose quality in the manuscript format.
Reviewer 1 comments:
Multi-Axial Analysis: It is unclear if the measurement instruments function solely on one axis. Expanding the analysis to include anteroposterior and lateral dimensions could enrich the understanding of squat movement dynamics and provide a more comprehensive perspective on athletic performance.
Note: The analysis is limited to a single axis! These are not triaxial measurement instruments. In the execution of a squat, its analysis, learning, and optimization, it could be crucial to be able to record anteroposterior and lateral values.
Author´s response: Done. No multi-axial analysis can be added since both devices present the net force values as the sum of the three axes. We have specified that in the “Material and Methods” section (lines 137-138).
Reviewer 1 comments:
Calibration of Instruments: Confirming the calibration of the measurement instruments would strengthen the validity of the results, especially under extreme load conditions.
p3 l 119-129. Have the measurement instruments been calibrated?
Author´s response: Done. Specifications about calibration of the instruments have been added in the “Material and Methods” section (lines 132-133).
Reviewer 1 comments:
Amplitude of Force Values: Discussing the amplitude of observed isometric force values would be useful to assess the consistency of measurements between the two systems under varying load conditions.
The document does not seem to mention the amplitude of the isometric force values developed by the subjects during the experiments. This would be necessary to verify if, under extreme load constraints, the comparison of values between the two systems remains coherent and statistically validated. It is possible that inconsistencies exist for these two systems for extreme minimal and/or maximal values.
Author´s response: Done. In relation to the amplitude of force values tested we have specified in a clearer manner that subjects performed maximum contractions (ln. 85, 106, 107). This was done to ensure real peak force values for every subject and every contraction since peak force is the main kinetic variable we want to test here. However, we recognize that including different percentages of this maximum in the analysis which will present a wider range of absolute force values could be of great relevance for testing reliability and validity. For that reason, we have added this issue as one more limitation (ln.355-357) and as one future line of research (ln. 372-373).
Reviewer 1 comments:
You could have tested the concordance of the values of the two measurement instruments through a static (isometric) maximum load constraint, using a system with a cable passing through a pulley, the other end being attached to the "GSTRENGTH" itself attached to the force platform.
Author´s response: Not suitable. The author find very interesting the possibility of testing maximum load capability of the devices (specially the strain gauge) implementing a methodology like the one suggested by the reviewer, however, such an approach will be very difficult since force plate are a pressure dependent device and will not function properly performing a direct pull on them (it is necessary to pull on a fixed surface while pushing against them as in our protocol). Additionally, is not possible to directly attach the strain gauge to the force plates. We consider that these issues will be resolved in part with the addition of the illustrations of the set-up as suggested by the reviewer (lines 134-136).
Reviewer 1 comments:
Please standardize the following references:
ABBOTT, B.C.; WILKIE, D.R.
"VALIDATION OF A COMMERCIALLY AVAILABLE STRAIN GAUGE AGAINST A SERIES OF KNOWN LOADS USING A SHORT TIME APPROACH.”
Author´s response: Done. References have been standardized (lines 397-398, 426-427).
Figure 3 (Figure 2 in previous version)

Figure 4 (Figure 3 in previous version)

Reviewer 2 Report
Comments and Suggestions for Authors
Dear authors,
your topic about "Validation of the Commercially Available Strain Gauge “GSTRENGTH” for Measuring Peak Force in the Isometric Belt Squat at Different Joint Angles" with the aim of "o test the reliability and validity of the “GSTRENGTH” for measuring PF in the isometric belt squat exercise" is very interesting and important for the literature of this topic.
In general the work is well performed, the manuscript is well written and has a good structure. I only have one suggestion that comes from your study limitation. The fact that you only have 5 subjects i cannot agree with the title of the article. I suggest to had a case study to the title or similar, add "Validation of the Commercially Available Strain Gauge “GSTRENGTH” for Measuring Peak Force in the Isometric Belt Squat at Different Joint Angles" is not suitable for the title given the limitations of your work.
congratulations
Author Response
REPLY TO REVIEWER 2
SENSORS
---------------------------
07 May 2024
Dear reviewer.
The authors of the manuscript want to thank you for your time and effort, as well as for your recommendations that for sure have helped us to improve the quality of our work.
In the following lines we specified the changes pertinent to the reviewer´s suggestions with the manuscript lines in which they should appear, in the manuscript changes are resalted in red letters to help the next revision. If the authors thinks that any suggestion is not suitable or appropriate a reasoned explanation will be presented.
Reviewer 2 comments:
Dear authors,
your topic about "Validation of the Commercially Available Strain Gauge “GSTRENGTH” for Measuring Peak Force in the Isometric Belt Squat at Different Joint Angles" with the aim of "o test the reliability and validity of the “GSTRENGTH” for measuring PF in the isometric belt squat exercise" is very interesting and important for the literature of this topic.
In general the work is well performed, the manuscript is well written and has a good structure. I only have one suggestion that comes from your study limitation. The fact that you only have 5 subjects i cannot agree with the title of the article. I suggest to had a case study to the title or similar, add "Validation of the Commercially Available Strain Gauge “GSTRENGTH” for Measuring Peak Force in the Isometric Belt Squat at Different Joint Angles" is not suitable for the title given the limitations of your work.
congratulations
Author´s response:
The authors want to thank again the reviewer for the recommendations for improving the present work. However, although the number of participants is very low, for our main objective the cases used for the analysis are the number of contractions (n = 120). For that reason, we consider that it will not be appropriate to use the term “case study” since the main results are based on the total sample of cases and not in a subject-by-subject way. This type of approach has been used previously in similar validation studies with even lower number of participants but with enough cases for analysis, two examples are:
N = 2 subjects, 96 cases
Balsalobre-Fernández, C., Agopyan, H., & Morin, J. B. (2017). The Validity and Reliability of an iPhone App for Measuring Running Mechanics. Journal of applied biomechanics, 33(3), 222–226. https://doi.org/10.1123/jab.2016-0104
N = 10 subjects, 150 cases
Balsalobre-Fernández, C., Kuzdub, M., Poveda-Ortiz, P., & Campo-Vecino, J. D. (2016). Validity and Reliability of the PUSH Wearable Device to Measure Movement Velocity During the Back Squat Exercise. Journal of strength and conditioning research, 30(7), 1968–1974. https://doi.org/10.1519/JSC.0000000000001284